

**Physical processes influencing the Asian climate due to black carbon emission**
**over East and South Asia**
*Feifei Luo[1], Bjørn H. Samset[2], Camilla W. Stjern[2], Manoj Joshi[3], Laura J. Wilcox[4],*
*Robert J. Allen[5], Wei Hua[1], and Shuanglin Li[6,7]*
1. School of Atmospheric Sciences/Plateau Atmosphere and Environment Key
Laboratory of Sichuan Province/Meteorological Disaster Prediction and Warning
Engineering Laboratory of Sichuan Province, Chengdu University of Information
Technology, Chengdu, China
2. CICERO Center for International Climate Research, Oslo, Norway
3. Climatic Research Unit, School of Environmental Sciences, University of East
Anglia, Norwich, United Kingdom
4. National Centre for Atmospheric Science, University of Reading, Reading, United
Kingdom
5. Department of Earth and Planetary Sciences, University of California Riverside,
Riverside, CA, United States of America
6. Climate Change Research Center, Institute of Atmospheric Physics, Chinese
Academy of Sciences, Beijing, China
7. Department of Atmospheric Science, China University of Geoscience, Wuhan,
China
Corresponding author: Feifei Luo (lff@cuit.edu.cn)



**Abstract**
Many studies have shown that black carbon (BC) aerosols over Asia have
significant impacts on regional climate, but with large diversities in intensity, spatial
distribution and physical mechanism of regional responses. In this study, we utilized a
set of Systematic Regional Aerosol Perturbations (SyRAP) using a reduced complexity
climate model, FORTE2, to investigate responses of the Asian climate to BC aerosols
over East Asia only, South Asia only, and both regions at once, and thoroughly examine
related physical processes. Results show that regional BC aerosols lead to a strong
surface cooling, air temperature warming in the low-level troposphere, and drying over
the perturbed areas, with seasonal differences in magnitude and spatial distribution.
Atmospheric energy budget analysis suggests that reductions in local precipitation
primarily depend on the substantial local atmospheric heating due to shortwave
absorption by BC. Increases in dry static energy (DSE) flux divergence partly offset the
reduced precipitation over north China in summer and most of China and India in the
other three seasons. Decreases in DSE flux divergence lead to stronger reduction in
precipitation over south China and central India in summer. Changes in DSE flux
divergence are mainly due to vertical motions driven by diabatic heating in the middle
and lower troposphere. BC perturbations also exert non-local climate impacts through
the changes in DSE flux divergence. This study provides a full chain of physical
processes of the local climate responses to the Asian BC increases, and gives some
insights to better understand the uncertainties of model responses.



## 1. Introduction

Black carbon (BC) aerosol, a short-lived pollutant and climate forcer, is emitted from the incomplete combustion of biomass and fossil fuels, and exerts significant effects on global and regional climate (Ramanathan and Carmichael, 2008; Bond et al., 2013; Stjern et al., 2017; IPCC, 2021; Li et al., 2022). Alongside rapid economic developments of China and India over the past few decades, East and South Asia have become the highest BC emissions hotspots in the world. Despite BC emissions from China decreasing substantially in the past decade, East and South Asia are expected to remain the highest BC loadings globally in the coming decades (Lund et al., 2019). Hence, the climate impacts of BC emissions from East and South Asia have been extensively investigated (e.g. Li et al., 2016; Lou et al., 2019; Xie et al., 2020; Westervelt et al., 2020; Herbert et al., 2022; Yang et al., 2022). Although many model studies have shown that Asian BC aerosols are of great importance for local climate (especially the Asian monsoonal systems), considerable uncertainty exits regarding the intensity and spatial distribution of the Asian climate responses to BC forcings, as well as the related physical mechanisms.

Menon et al. (2002) found that during summer, a large BC forcing induced a "southern flooding and northern drought" (SFND) precipitation pattern in China, and moderate cooling in China and India based on the Goddard Institute for Space Studies (GISS) global climate model (CGCM). However, many subsequent studies were not able to reproduce the SFND pattern, in part due to the poor representation of the Asian summer monsoon in models (Wilcox et al., 2015). Gu et al. (2006) found that BC forcing acted to suppress precipitation in southern and eastern China due to warming in the middle to high latitudes based on an atmospheric general circulation model (AGCM). Zhang et al. (2009) showed that the effects of direct radiative forcing due to increased BC aerosols can decrease precipitation and increase surface temperature in southern China and India, but cause the opposite responses in northern China, by comparing CAM3 AGCM simulations with the all-aerosol types and without carbonaceous aerosol. Liu et al. (2018) found a similar dipole precipitation pattern with a decrease in southern China and an increase in the north mainly due to the fast





adjustments (simulations with fixed SST) of BC forcing, and a surface cooling over
Asia except over the Himalayan region in the 10x modern Asian BC emissions or
concentrations experiments in the Precipitation Driver Response Model
Intercomparison Project (PDRMIP) (Myhre et al., 2017).

Wang et al. (2017) conducted CESM1 simulations with increased BC emissions

from preindustrial to present-day level, and proposed that the fast adjustment
strengthened the EASM, while slow adjustment (response to aerosol-induced SST
change) dominated the spatial pattern of precipitation response, which showed a tripolar
precipitation pattern with wetting-drying-wetting from north to south China, although
the responses in most regions were statistically insignificant. The fast/slow responses
in Wang et al. (2017) are different with those in Liu et al. (2018). Xie et al. (2020) used
the PDRMIP ensemble to show that the extremely high global/Asian BC forcing can
cause the similar tripolar precipitation pattern over eastern China in summer, and they
stressed that these responses mainly result from the enhanced upper-level atmospheric
temperature over Asia instead of low-level thermal feedbacks. Mahmood and Li (2014)
showed that South Asian BC also can induce the tripolar precipitation pattern over East
China via a propagating wave train along the Asian upper tropospheric jet based on
ensemble sensitive experiments in GFDL AM2.1.

Persad et al. (2017), based on GFDL AM3, have shown that surface solar dimming

dominates the reduction of the East Asian summer precipitation due to the decreased
land-sea contrast, whilst atmospheric heating from absorbing aerosols partially offset
the reduction, which leads to a smaller reduction when both effects are considered
simultaneously. Jiang et al. (2013) and Guo et al. (2013) indicated that there was no
statistically significant change in precipitation over East China in response to global
BC forcing during summertime, based on the CAM5 and HiGAM AGCM experiments,
respectively. Although there are many differences across studies, some similarities can
still be found in most of studies, such as surface cooling in the perturbation area, wet
response in North China.

In addition, some AGCM/CGCM studies suggested that increased BC aerosols can

lead to a weakened South Asian summer monsoon (SASM) (Lau and Kim, 2007; Meehl



et al., 2008), while others found that there should be a strengthened SASM via
increasing the atmospheric meridional land-sea thermal gradient, or an elevated heat
pump effect (Lau and Kim et al., 2006; Xie et al., 2020; Westervelt et al., 2020). On the
other hand, Westervelt et al. (2018) conducted aerosol removal experiments using three
CGCMs, and found that Indian BC decreases lead to no change or a small decrease in
precipitation in India.

By comparing the sum of Asian summer climate responses to individual responses

over East and South Asia with the responses to simultaneous forcing, the regional
linearity of BC forcing has been investigated. Chen et al. (2020) and Herbert et al. (2022)
suggested that the responses were highly nonlinear due to the interaction of atmospheric
circulation changes, based on the regional climate model RegCM4 and the Intermediate
General Circulation Model 4 (IGCM4) simulations, respectively. In contrast, Reccia
and Lucarini (2023) and Stjern et al. (2024) found that the responses were almost linear
in most of Asian regions. The difference may be related to the different spatial extend
of the aerosol perturbation in the simulation design (Stjern et al., 2024).

For the East Asian winter climate, Jiang et al. (2017) found that BC forcing can

lead to an intensification of the East Asian winter monsoon (EAWM) northern mode
via heating Tibetan Plateau using the CAM5 model. On the contrary, Lou et al. (2019)
suggested that BC emitted from the North China can weaken the EAWM through ocean,
sea ice, cloud feedbacks based on CESM. BC aerosol also can impact on spring and
autumn precipitation in China (Guo et al., 2013; Hu and Liu et al., 2014; Deng et al.,

2014).

These inconsistent results, and the large uncertainty in the simulated response of

the Asian climate to BC changes, are partly related to differences in the modeling
approach (e.g., AGCMs versus CGCMs/ESMs) and also the magnitude and location of
the BC perturbation. Atmosphere-only GCMs lack SST feedbacks, which are crucial in
influencing the Asian monsoon (Dong et al., 2019), while CGCMs or ESMs involving
more and complex physical processes make it difficult to identify the key physical
processes of impacts of regional BC aerosols on Asian climate. The inconsistency may
also be associated with model-specific differences. Different models that include



different physical processes, combined with different experiment designs that can
influence the atmospheric circulation response, mean that understanding the causes of
differences between studies is very difficult. Hence, reduced complexity models, such
as FORTE2 (Fast Ocean Rapid Troposphere Experiment version 2), provide an
alternative and useful tool for such studies, given that such models not only include all
the main mechanisms of aerosol-climate interactions, but also allow fast speed of
integration and longer simulations with lower cost. Stjern et al. (2024) have utilized
FORTE2 to perform a series of Systematic Regional Aerosol Perturbations (SyRAP)
simulations with absorbing and scattering aerosol species in East Asia only, South Asia
only, and both regions simultaneously. Their results have shown that SyRAP-FORTE2
is a helpful framework to understand and decompose the local and remote climate
effects of regional aerosol emissions.
Therefore, based on the simulations of the regional BC perturbations in SyRAP-
FORTE2, this study aims to address the following two questions: (1) what are responses
of Asian climate to either East Asian or South Asian BC emissions, or both regions at
once, respectively? (2) What are key physical processes involved in these responses?
The rest of the paper is organized as follows: section 2 describes the FORTE2
model, the SyRAP simulations, and the analysis methods and datasets; section 3 briefly
evaluates the climatology of SyRAP simulations, examines responses to the regional
BC perturbations in SyRAP-FORTE2, and investigates the underlying physical
processes involved in these responses; section 4 compares the results of the atmospheric
energy budget in SyRAP-FORTE2 with those in the PDRMIP models; Finally, the
summary and discussion are provided in section 5.
**2.  Methods**
**2.1 The FORTE2 model**
FORTE2 is an intermediate-complexity coupled atmosphere-ocean general
circulation model (Blaker et al., 2021). The atmospheric component is the Intermediate
General Circulation Model 4 (IGCM4) with a horizontal resolution of approximately
2.8° (T42), and 35 sigma levels extending up to 0.1 hPa (Joshi et al., 2015). IGCM4



includes schemes for radiation, land-surface properties, convection, precipitation, and
clouds (Zhong and Haigh, 1995; Betts and Miller, 1993). The oceanic component is the
Modular Ocean Model-Array (MOMA) with a horizontal grid spacing 2°×2°, and 15 z-
layer levels increasing in thickness with depth from 30m at the surface to 800m at the
bottom (Webb, 1996). Sea ice is represented by a barrier to heat fluxes between the
ocean and atmosphere components. FORTE2 runs without flux adjustments (Blaker et
al., 2021). Blaker et al. (2021) have thoroughly evaluated the skill of FORTE2 in
simulating the atmosphere, ocean and major climatic modes. Hence, given the
advantages of FORTE2 in terms of running speed, flexibility and economy, it is a useful
tool to study a wide range of climate questions. For the SyRAP-FORTE2 simulations,
FORTE2 was updated to include a parameterization of aerosol-cloud interactions,
where cloud droplet effective radius in low- and mid-level clouds is reduced from 15μm
to 10μm in regions where optical depth is greater than 0.07. However, this
parameterization is only applied for scattering aerosol, and is not used in the BC
experiments described in the following section. The semi-direct effect of BC is included.
**2.2 The SyRAP-FORTE2 simulations**

Stjern et al. (2024) performed a set of SyRAP experiments using the FORTE2

model. The main simulations include: (1) baseline simulations forced by GHG
concentrations at different climate states (i.e. preindustrial, present-day and future $CO_2$
levels) and no aerosol; (2) perturbation simulations forced by added absorbing (BC, and
organic carbon, OC) or scattering (sulfate, $SO_4$) aerosols over East Asia only, South
Asia only, and over both regions simultaneously with only aerosol-radiation
interactions (ARI), and GHG concentrations at different climate states; (3) Aerosol-
cloud interactions (ACI) simulations forced by added $SO_4$ in the combined East Asia
and South Asia region in which the ACI were turned on. This study focusses on the
effects of adding regional BC perturbations, but the SyRAP-FORTE2 design allows the
impacts of different aerosol species to be compared in a consistent framework.

FORTE2 did not use aerosol gas emissions/concentrations as most CMIP6 models

use. Instead, the global gridded monthly aerosol optical depths (AOD) and vertical





distributions were used from the Copernicus Atmosphere Monitoring Service reanalysis
(CAMSRA) during 2003-2021 (Inness et al., 2019). Aerosols are not transported in
FORTE2. The application of aerosol distributions from a reanalysis means that the
simulations include a realistic aerosol spatial distribution, but the lack of aerosol
transport means that there are no feedbacks between the climate response and the
aerosol distribution (e.g. increased precipitation leading to increased aerosol removal).
All simulations were run for 200 years, with years 51-200 used for analysis.

In this study the baseline simulation (piC) and the BC perturbation simulations of

three different regions at the pre-industrial climate conditions (280 ppmv) were
performed to explore potential physical processes of Asian BC aerosols influencing the
local climate (Table 1). There is no significant difference in Asian climate responses to
BC aerosols at different background climate states in the SyRAP-FORTE2 simulations
(Stjern et al., 2024). The regional annual mean BC AOD perturbation is about 0.015 for
East China, and about 0.01 for India, respectively (Figure 1a in Stjern et al., 2024). Note
that only climate impacts due to ARI were considered here.

### 2.3 Analysis methods and Datasets

**2.3 Analysis methods and Datasets**

The response to a particular regional forcing is estimated by the mean difference

between the perturbation simulation and the baseline simulation. Statistical significance
of the response is assessed using a two-tailed Student's t-test.

The atmospheric energy budget is applied to understand the precipitation

responses (Muller and O'Gorman, 2011; Richardson et al., 2016; Liu et al., 2018). The
energy associated with precipitation can be separated into a thermodynamic component
with only changes in the diabatic cooling (Q), and a dynamic component with only
changes in the dry static energy (DSE) flux divergence (H), as shown in Eq. (1):
$$L_c\delta P = \delta Q + \delta H \quad (1)$$
where $L_c$ is the latent heat of condensation, P is precipitation, δ denotes a perturbation.
Then,
$$\delta Q = \delta LWC - \delta SWA - \delta SH \quad (2)$$
where LWC is atmospheric longwave cooling, SWA is atmospheric shortwave



absorption, and SH is sensible heat flux from the surface.

$\delta H$ is calculated as a residual between $L_c \delta P$ and $\delta Q$. Furthermore, H can be seen

as the sum of the changes in mean ($H_m$) and eddy ($H_{trans}$) components. $\delta H_m$ can be
decomposed into four components associated with dynamic and thermodynamic effects
on vertical and horizontal advection of DSE, as shown in Eq. (3):
$$\delta H_m = \delta H_{Dyn\_v} + \delta H_{Thermo\_v} + \delta H_{Dyn\_h} + \delta H_{Thermo\_h}$$
$$= \int \delta\overline{\omega}\frac{\partial\overline{s}}{\partial p}\frac{dp}{g} + \int \overline{\omega}\delta(\frac{\partial\overline{s}}{\partial p})\frac{dp}{g} + \int \delta\overline{u} \bullet \nabla\overline{s}\frac{dp}{g} + \int \overline{u} \bullet \delta(\nabla\overline{s})\frac{dp}{g} \ (3)$$
where $\omega$ is vertical velocity, $s$ is DSE, $p$ is pressure, $g$ is the gravitational acceleration,
$u$ is horizontal wind vector, $\nabla$ is the horizontal gradient, and an overbar indicates
climatological monthly means. Therefore, $H_{Dyn\_v}$ is related to changes in vertical
velocity, $H_{Thermo\_v}$ is related to changes in vertical DSE gradients, $H_{Dyn\_h}$ is related to
changes in horizontal winds, and $H_{Thermo\_h}$ is related to changes in horizontal DSE
gradients. $\delta H_{trans}$ is calculated as a residual between $\delta H$ and $\delta H_m$.

To evaluate the ability of FORTE2 to represent the observed climate monthly

precipitation, surface temperature (Ts), and horizontal wind components were used
from the NOAA-CIRES-DOE 20th Century Reanalysis V3 (20CR) on a 2° × 2° grid
spanning 1836 to 2015 (Compo et al., 2011). The 20CR is only based on surface
observations of synoptic pressure of NOAA's physical Sciences Laboratory. Monthly
sea level pressure (SLP) data were from the Hadley Centre (HadSLP) in a horizontal
resolution of 5° × 5° (Allan & Ansell, 2006).

To compare the response of FORTE2 to BC perturbations to the responses from

CMIP class models we used the PDRMIP 10 times the modern Asian BC
concentrations/emissions perturbation simulations and the baseline simulations with
modern aerosol concentrations/emissions and greenhouse gases with the year 2000,
based on the 5 GCMs (CESM1-CAM5, GISS-E2-R, HadGEM3, MIROC, and
NorESM1; Table S1). More details of the PDRMIP design can be found in Myhre et al.
(2017), while an overview of the monsoon response is given in Liu et al. (2018).
**3. Results**





### 3.1 Evaluation of baseline climate in SyRAP-FORTE2

Blaker et al. (2021) present a detailed overview of the FORTE2 climatology. The
model simulates the Asian climate well, and the more focused evaluation presented by
Stjern et al. (2024) demonstrated that FORTE2 is an appropriate tool to study the Asian
climate effects of local aerosol perturbations. Here, for completeness, we present an
overview of the skill of FORTE2 in simulating the Asian climate. The seasonal
evolutions of precipitation and surface temperature (Ts) in East China and India, and
the climatology of lower tropospheric circulation (SLP and 850 hPa horizontal wind)
in the baseline experiment (piC) are compared with those in the reanalyses from 1851
to 1896. FORTE2 reproduces the seasonality of precipitation in East China reasonably
well, but slightly overestimates the averaged magnitude with about 1 mm/day in
summer (Fig. 1a). However, the model underestimates the magnitude of South Asian
Summer Monsoon precipitation by approximately 4 mm/day during June-September
relative to the 20CR reanalysis. Dry biases in Indian summer precipitation of similar
magnitudes are found in many CMIP5 and CMIP6 models (Sperber et al., 2013; Wilcox
et al., 2020; Liu et al., 2024). Differences between reanalyses and observational datasets
can also have similar magnitudes (Wilcox et al., 2020). Hence, we conclude that the
FORTE2 representation of monsoon precipitation is suitable for our study. The model
performs fairly well in the seasonality of temperature and the magnitudes in both
regions (Fig. 1b).
The simulated SLPs are generally lower than that in HadSLP in all four seasons,
especially the Siberian High in winter, the western North Pacific Subtropical High
(WPSH) and the Indian Low in summer (Fig. 1c-j). Compared to the reanalyses, the
simulated Indian Low is too strong, and its eastern fringe and the westerly from the
Indian Ocean extend too far east into the western North Pacific, so does the Indian
summer monsoon trough (Fig. 1g and h). This corresponds the model dry bias over the
Indian subcontinent (Fig. 1a and S1e-f). Meanwhile, the WPSH is too weak to expand
sufficiently far west, corresponding to the very weak easterly along the southern fringe
of the WPSH, which lead to relatively less rainfall over the Philippines (Fig. S1e-f) and
underestimate of the effects of the WPSH on the East Asian summer monsoon (Fig. 1g-



h). Despite these deficiencies, the model captures the essential features of lower
tropospheric circulation, precipitation and temperature over Asia, which is consistent
with Stjern et al. (2024).
**3.2 Temperature and Precipitation responses**

Figure 2 shows spatial patterns of Ts responses to increased Asian BC aerosols in

four seasons. Firstly, there is a substantial land cooling over the perturbed regions in all
four seasons, but with distinct seasonal differences in distributions and values. Under
BC_CHI, a cooling can be seen in most of China in winter and spring (Fig. 2a-b), with
the area-averaged values of -0.9±1.2 K (mean value ± 1 standard deviation) and -
1.1±0.9 K, respectively (Fig. 3a). The large standard deviations indicate the large spread
of distributions of responses (Fig. S2). There is a slight warming of sea surface
temperatures near China in spring.  In summer and autumn, a cooling is seen mainly in
the region to the north of the Yangtze River valley, especially in North China, while
there is a weak warming to the south (Fig. 2c-d). The area-averaged values therefore
are relatively smaller than those in winter and spring, showing about -0.7±0.7 K for
summer and autumn (Fig. 3a).

Under BC_IND, the strongest cooling occurs in the whole India in spring with the

area-averaged value of -1.4±1.4 K, but a weak warming in the tropical Indian Ocean
(Fig. 2f and 3b). The significant cooling is concentrated in northern India in the other
three seasons, with the area-averaged Ts decreased by 0.9±1.3 K for winter and autumn,
0.6±1.1 K for summer (Fig. 2e, g, h and 3b). The ocean surrounding the Indian
subcontinent shows a smaller cooling in the three seasons.

Secondly, the BC perturbations not only cause the local responses, but also non-

local responses through atmospheric circulations. The increased BC over East China
leads to a dipolar pattern in India with a warming to the north and a cooling to the south
in summer and autumn, and a significant warming in Central Asia in summer and
Southeast Asia in autumn (Fig. 2c-d). BC emissions over India can induce a cooling in
Southeast Asia in all seasons, and in summer the responses are the strongest with the
averaged value of about 1.15 K. Meanwhile, there is a significant cooling in central and



southern China in summer and autumn, with decreases of 0.3±0.7 K and 0.2±0.7 K,
respectively (Fig. 2g-h and 3a). These added impacts lead to an East Asian cooling
response under BC_CHI+IND that is stronger than that under BC_CHI in summer and
autumn (Fig. 2k-l and 3a). Moreover, by comparing the responses to both regions at
once to the sum of responses to the two separate regions, it is found that the differences
are almost insignificant, suggesting that the impact of Asian BC aerosol on the Asian
Ts is linear regionally in all four seasons (Fig. S3).

In contrast to Ts responses, the responses in air temperature (Ta) at 850 hPa show

a significant warming over much of the perturbed regions (Fig. 3c-d and 4). A cooling
occurs over a few perturbed regions, such as central China under BC_CHI and
BC_CHI+IND in winter (Fig. 4a, i). The vertical distribution of the temperature
response is characterized by cooling at the surface and warming in the lower
troposphere. This is a result of atmospheric absorption of solar shortwave radiation (SW)
by BC aerosols, and is consistent with many previous studies (e.g., Li et al., 2016).
Outside the perturbed regions, the Ta responses are in line with the Ts response, for
example a cooling over China and Southeast Asia under BC_IND in summer and
autumn (Fig. 3c and 4g, h).

Increased BC over East China primarily induces robust drying over most of China

in all four seasons (Fig. 3e and 5a-d). Specifically, the spatial distribution in summer
shows a substantial decrease over southern China (up to 40%), a relatively weaker
decrease over north China and a weak increase over northwest China, with the area-
averaged precipitation decreased by 0.5±0.5 mm/day (Fig. 3e, 5c and S4c). In spring
and autumn, the decreased precipitation is mainly located in south and northeast China
by up to about 40% (Fig. 5b and d). Although the absolute change is the weakest in
winter (-0.2±0.3 mm/day), the relative change in central China decreases by up to about
60% (Fig. 3e, 5a and S4a). Under BC_IND, the response over the Indian subcontinent
is only statistically significant in summer, with a decrease of 0.5±1.1 mm/day (~21%)
(Fig. 3f and 5g).

Additionally, under BC_CHI there is increased rainfall over India in summer and

autumn, which can counteract the local decrease due to Indian BC forcing (Fig. 3f and





5c-d). Hence, when considering the simultaneous BC forcing in the two regions
(BC_CHI+IND) in summer and autumn, there is no significant change in the regional
mean, showing a dipolar pattern with more rainfall over northern India and less rainfall
over southern India (Fig. 3f and 5k-l). On the other hand, BC_IND can induce a
significant precipitation increase over China and Southeast Asia in spring and summer
(Fig. 5f-g), and partly offset the decrease in response to BC_CHI over China. Hence,
there are weaker responses under BC_CHI+IND for China than those under BC_CHI
(Fig. 3e). For regional linearity, the responses are almost linear (Fig. S5). Only few
regions exhibit nonlinear response, such as a decrease over south Thailand in winter, an
increase over northeast India in spring and summer. The above analysis regarding
summer part has been included in Stjern et al. (2024), but is included here as a precursor
to the detailed mechanistic analysis that follows.
**3.3 Energy balance response**
The following analysis mainly focuses on summer for clarity; the results in the
other three seasons are in the supplement (Figures S7-S10, S12-S16). The spatial
patterns of net TOA and surface energy responses to regional BC aerosols in summer
are illustrated in figure 6. As expected, there are increases in net downward TOA
shortwave (SW) with area mean responses of 7.3~9.6 W/m$^2$ associated with decreases
in convective clouds over the perturbed regions (Fig. 6a-c and S6j-l). Decreases in net
surface downward SW can be seen with area mean responses of -22.6~-27.8 W/m$^2$ due
to SW absorption by BC aerosols (Fig. 6d-f). Significant increases in low and middle
clouds also contribute to the reduction in surface SW (Fig. S6d-i). Hence, there is
warming in the troposphere and cooling at surface in these perturbed regions. The
enhanced net surface upward longwave (LW) has a small contribution to the local
surface cooling with area mean responses from -3.2 to -5.8 W/m$^2$ (Fig. 6g-i), which is
related to the decreases in convective clouds (Fig. S6j-l). On the contrary, the positive
changes in downward sensible heat (SH) (8.1~9.8 W/m$^2$) and latent heat (LH) (9.5~12.9
W/m$^2$) cause a warming effect partly offsetting the cooling in the perturbation regions
(Fig. 6j-o). The decreased SH is caused by the vertical temperature differences between



the surface and lower atmosphere, and the decreased LH associated with heating aloft
and precipitation. Outside the perturbation regions, the negative changes in SW, LW
and LH are responsible for the cooling in Southeast Asia and south China induced by
the increased BC in India (Fig. 6e, h, n). Similar results can be seen in other three
seasons (Table S2).
Figure 7 shows area-averaged atmospheric column energy budget terms (see Eq.
1) over East China and India in summer. For the drying in East China/India under
BC_CHI/BC_IND, the substantial reductions of the diabatic cooling ($\delta Q$) are the prime
driver of the decreases in the energy of precipitation ($L_c\delta P$), while the increases in the
DSE flux divergence ($\delta H$) offset the effect of $\delta Q$ to a large extent.
The enhanced precipitation in China under BC_IND, and in India under BC_CHI,
is the result of increased $\delta H$ (blue and green bars in Fig. 7 are almost identical), with
negligible contribution from $\delta Q$. Hence, the local precipitation response to local BC
increases is largely driven by $\delta Q$, while the remote precipitation response is largely
driven by $\delta H$. Consequently, it can be found that the larger changes in $\delta H$ lead to the
smaller responses in $L_c\delta P$ over China and India under BC_CHI+IND, relative to the
changes under BC_CHI and BC_IND. It should be mentioned that the relatively large
uncertainties (error bars in Fig. 7) of $L_c\delta P$ mainly depend on $\delta H$. The other three
seasons show the similar results (Fig. S7).
Spatially, $\delta Q$ shows a significant decrease throughout the entire perturbed regions
in the three simulations, with large decreases located over North China and northern
India (Fig. 8d-f). Under BC_CHI, $\delta H$ is characterized by a dipolar pattern with positive
changes in north of the Yangtze River and negative in south (Fig. 8g). Hence, $\delta H$ and
$\delta Q$ cancel each other out in North and Northeast China, and combine with each other
in south of the Yangtze River. As a result, there is the strongest decrease in precipitation
in south China, and relatively weak decrease in north China (Fig. 8a). Under BC_IND,
significant positive changes in $\delta H$ can be found along the southern edge of Himalayas
and the southern tip of the Indian subcontinent, and weak and nonsignificant negative
changes in central India (Fig. 8h). $\delta H$ can therefore offset the negative changes in $\delta Q$
in north and south India. The substantial $L_c\delta P$ reduction is concentrated in central India



mainly due to δQ (Fig. 8b).

In addition, δH is the dominating factor for $L_c\delta P$ beyond the perturbation regions,

for example positive changes in northeast India under BC_CHI, positive changes from
Southeast Asia to the tropical western Pacific and central China under BC_IND (Fig.
8g-h). Under BC_CHI+IND, it can be found that the extent and magnitude of δH are
larger than those in the simulations of individual regions (Fig. 8i), which indicates more
balance between δH and δQ, corresponding to relatively weaker precipitation over East
China and India (Fig. 7 and 8c). Relative to summer, δQ and δH has negative and
positive changes in the whole perturbation regions in the other three seasons,
respectively (Fig. S8-10). In winter and spring, there is a marked seesaw pattern of δH
between Asia and the tropical Indian Ocean and maritime continent under
BC_CHI+IND, leading to less precipitation in the latter regions (Fig. S8c, i and S9c, i).
Overall, the combined effect of δH and δQ shape the spatial pattern of precipitation
responses to the regional BC.

The reductions in δQ (see Eq. 2) are dominated by the strong atmospheric heating

due to SW absorption by BC aerosols ($-\delta SWA$), and also contributed by the small
decreases in the atmospheric longwave cooling (δLWC) (Fig. S11a-f). The sensible heat
flux from the surface ($-\delta SH$) plays a role in increasing δQ, but with relatively small
values (Fig. S11g-i).
**3.4 Dynamic processes responsible for responses**

Due to storage constraints, 3D atmospheric output from FORTE2 was archived on

three pressure levels, 250 hPa, 500 hPa, and 850hPa, to capture three key aspects of the
tropospheric circulation response. While this precludes a quantitative analysis of the
component terms of $\delta H_m$ (Eq. 3), it is sufficient to identify to main contributing term.
We now examine the four terms of $\delta H_m$ (see Eq. 3), including the dynamic components
with changes in vertical and horizontal atmospheric circulations ($\delta H_{Dyn\_v}$ and $\delta H_{Dyn\_h}$),
and thermodynamic components with changes in vertical and horizontal DSE gradients
($\delta H_{Thermo\_v}$ and $\delta H_{Thermo\_h}$). Figure 9 displays spatial patterns of the four components



in summer in the three simulations. In general, the $\delta H_{Dyn\_v}$ highly resembles the $\delta H$ in
the three simulations, and the magnitudes in $\delta H_{Dyn\_v}$ are far greater than those in the
other three terms, suggesting that dynamic effect of vertical circulation is the primary
contributor to $\delta H$ (Fig. 9a-c). $\delta H_{Thermo\_v}$ and $\delta H_{Dyn\_h}$ are small in all regions for all
experiments. Larger anomalies are seen in $\delta H_{Thermo\_h}$, where negative anomalies offset
some of the influence of $\delta H_{Dyn\_v}$ over the Indo-China peninsula in the BC_IND and
BC_CHI+IND experiments. However, these anomalies are not sufficient to influence
the sign of $\delta H$, which is still primarily driven by $\delta H_{Dyn\_v}$ in this region. In the other
seasons, $\delta H_{Dyn\_v}$ remains the most important factor (Fig. S12-14), although it is more
strongly offset by $\delta H_{Dyn\_h}$ and $\delta H_{Thermo\_h}$ in winter. The effects of horizontal
circulation are relatively weak in spring and autumn.

Based on the above analysis, we conclude that vertical movement is the most

important contributor to $\delta H_m$. As expected, the spatial patterns of responses in Omega
(vertical velocity) at 500 hPa correspond well to those in $\delta H_{Dyn\_v}$ (Fig. 10a-c).
Anomalous ascent corresponds to the increase in $\delta H_{Dyn\_v}$, leading to more precipitation,
which offsets the precipitation reduction driven by decreased $\delta Q$. Anomalous descent
suppresses precipitation, adding to the precipitation reduction driven by the reduction
in $\delta Q$.

Why does the vertical velocity exhibit such changes? It seems to be related to the

temperature responses in the troposphere, reflected by a good corresponding
relationship between the Omega and Ta responses at 500 hPa (Fig. 10a-f). The warm
anomalies favor a divergence in the middle troposphere, which in turn are associated
with anomalous ascent. The cold anomalies are associated with a convergence and
descending motion. The above-mentioned relationship also exists in the other seasons,
and it is more pronounced at 850 hPa (Fig. 4 and S15).

From the transects of zonal mean diabatic heating over the perturbation regions,

changes to tropospheric heating can be seen more clearly (Fig. 11). Under BC_CHI, the



responses in diabatic heating show a meridional dipolar structure through the whole
troposphere over East China, with a cooling over the region to south of 32°N (the
Yangtze River basin), and a warming to north (Fig. 11a). The dipolar pattern
corresponds well to the meridional distributions of precipitation, vertical velocity and
Ta at 500 hPa. The cooling center located at the middle troposphere is due to the reduced
latent heat release caused by the substantial decrease in precipitation over south China.
The heating center at the lower troposphere in the north mainly results from SW
absorption by increased BC aerosol. The difference between south China and north
China is associated with the larger AOD perturbation imposed north of the Yangtze
basin in SyRAP-FORTE2 (Stjern et al., 2024). In BC_CHI+IND, there is a similar
dipolar pattern, except for a warming at the lower troposphere around south of 30°N
(Fig. 11b). The warming is related to the increased precipitation over south China
because of the BC aerosols over India.

For India in BC_IND, there is also a cooling center associated with the reduced

precipitation in the middle troposphere (Fig. 11c), corresponding to a cold anomaly and
descending motion at 500 hPa (Fig. 10b, e). In the lower troposphere, a warming can
be seen at south of the Qinghai-Tibet Plateau. Compared with BC_IND, the cold
anomaly is weaker, but the warm anomalies are strengthened under BC_CHI+IND (Fig.
11d). Hence, a significant ascending motion can be found in northeast India resulted
from the effect of increased BC over East Asia (Fig. 10c), which is consistent with that
in Herbert et al. (2022). In the other three seasons, however, unlike in summer, there is
no cooling center in the middle troposphere, and the heating centers are situated at the
lower troposphere (Fig. S16). Overall, the diabatic heating induced by the increased BC
aerosols at the lower troposphere leads to an ascending motion explaining the increased
δH over the perturbation regions.

Considering that the dynamic and thermodynamic effects of horizontal

atmospheric circulations have some contributions to δH, we look at the changes in
lower tropospheric horizontal circulation in response to changes of regional BC
aerosols (Fig. 10g-i). Under BC_CHI, the cyclone anomaly over East China leads to
anomalous easterly wind over North China with cutting off the moisture supply from





south (Fig. 10g). Under BC_IND, the westerly anomalies associated with the cyclonic
circulation over India favor to strength the Indian summer monsoon, which corresponds
to the increase in $\delta H_{Dyn\_h}$ (Fig. 10h). The responses in the circulations to both regions
at once can be seen as the sum of responses to the two separate regions (Fig. 10i).
Additionally, there is a cyclonic circulation over East China and an anticyclonic
circulation over central China in winter under BC_CHI (Fig. S17), leading to
anomalous northerly wind across central China and then suppress precipitation over
there, which is in agreement with the decrease in $\delta H_{Dyn\_h}$ (Fig. S10g). The changes in
horizontal circulations are related to the changes in Ta and omega in the lower
troposphere (Fig. 4 and S15).
**4    Energy budget analysis in other coupled models**
To evaluate the precipitation response and the mechanisms in FORTE2, we
compare the results of energy budget analysis (see Eq. 1) in the PDRMIP simulations
forced by 10 times the present-day Asian BC concentrations/emissions in five CMIP-
class models to those in the SyRAP-FORTE2 BC_CHI+IND experiment. Spatial
patterns of summer energy budgets in the PDRMIP models are illustrated in Figure 12.
There are significant decreases in $\delta Q$ over most of Asia in all of the PDRMIP models,
which is generally consistent with the results under BC_CHI+IND (Fig. 12f-j). Three
models (CESM1-CAM5, GISS-E2-R and NorESM1) have similar distributions of $\delta Q$
to BC_CHI+IND, showing a maximum center in North China and northern India.  $\delta H$
increases significantly in India and most of East China in these models (Fig. 12k, l, o),
again roughly resembling the changes of BC_CHI+IND (Fig. 8i), while the other two
models show a significant increase in north India and North China with weaker
magnitudes (Fig. 12m-n). The PDRMIP multi-model mean changes in $\delta H$ [figure 7 in
Liu et al. (2018)] are also similar to the changes in BC_CHI+IND. In spite of these
model differences in the individual terms, models are broadly consistent in their total
$L_c\delta P$ responses. Precipitation generally increases over India and North China, and
decrease over South China in the PDRMIP models and FORTE2 (Fig. 12a-e). There is



no significant increase in δH and precipitation in Southeast Asia in the PDRMIP models.

Figure 13 shows the regional means of each energy budget term over East China

and India in summer under the PDRMIP models. There is a weak and insignificant
reduction in $L_c\delta P$ from -0.6 (HadGEM3) to -7.2 W/m$^2$ (NorESM1) for East China,
which are comparable to the value of -3.8 W/m$^2$ under BC_CHI+IND (Fig. 13a). Hence,
the result under BC_CHI+IND in FORTE2 is agreement with the PDRMIP models,
suggesting that the increased BC perturbations over Asia lead to slight decreases in
precipitation over East China in all of the models. The regional means range from -1.2
(MIROC) to 13.7 W/m$^2$ (NorESM1) for India (Fig. 13b). Except for MIROC, $L_c\delta P$ in
the other models have stronger increases than that under BC_CHI+IND, about 0.6
W/m$^2$, which may be related to the larger drying bias of Indian summer precipitation in
FORTE2.

The effect of δQ decreases $L_c\delta P$ with a large range from -4.7 (HadGEM3) to -29.8

W/m$^2$ (GISS-E2-R) for East China, while δH has an opposite effect from 2.1 (MIROC) to
24.4 W/m$^2$ (GISS-E2-R) (Fig. 13a). Similarly, δQ changes from -5.7 (HadGEM3) to
-25.8 W/m$^2$ (GISS-E2-R) for India, and δH from 4.9 (MIROC) to 36.5 W/m$^2$ (GISS-
E2-R) (Fig. 13b). The magnitudes in δQ and δH under the PDRMIP models are much
smaller than those in FORTE2 in both the two regions, except for GISS-E2-R. The
negative effect of δQ and the positive effect of δH also can be seen in the PDRMIP
multimodel mean for the whole Asian region (Liu et al., 2018). Despite large difference
in the magnitudes of their responses, which are to be expected from their large range of
aerosol radiative forcing and climatological precipitation, the results of these models
are overall consistent qualitatively.

**5   Conclusion and Discussion**

In this study, we have investigated the Asian climatic responses to adding BC

aerosols to the separate regions (East China and India), and both regions at once, and
examined the associated physical processes, with the SyRAP simulations based on the
reduced-complexity climate model FORTE2. Our main findings are as follows.

i.    BC increases over East Asia or South Asia lead to a local strong surface cooling



540 and lower tropospheric air temperature warming in all four seasons, with seasonal

541 differences in magnitude and spatial distribution. The responses in temperature are

542 dominated by the substantial decreases in surface SW radiation due to SW absorption

543 by BC aerosols. BC over East Asia causes significant drying in south and northeast

544 China in spring, summer and autumn. In winter, there is a significant reduction in

545 central China. BC over South Asia induces a substantial decrease in rainfall in India in

546 summer. Also, South Asian BC induces significant decreases in temperature and

547 precipitation in Southeast Asia in summer and autumn.

548 ii. Responses in temperature and precipitation to Asian BC forcing are mostly

549 linear regionally in all four seasons. There are relatively smaller decreases in

550 precipitation responses to adding BC over both regions simultaneously, compared to

551 the local reductions in precipitation responses to BC increases over East Asia and South

552 Asia separately. This is because BC over East Asia (BC over India) increases

553 precipitation in northeast India, while BC over South Asia increases precipitation over

554 southern and central China.

555 iii. Using an energy budget analysis, we find that reductions in the energy of local

556 precipitation ($L_c\delta P$) over the perturbation regions result from decreases in net

557 atmospheric diabatic cooling ($\delta Q$). The increases in the dry static energy (DSE) flux

558 divergence ($\delta H$) play a role in offsetting the effects of $\delta Q$ to a large extent.

559 Consequently, the responses in precipitation to Asian BC can be considered as the result

560 of interactions between thermodynamic and dynamic processes. For $\delta Q$, the reductions

561 are mainly due to the strong atmospheric heating ($-\delta SWA$). For $\delta H$, the increases

562 depend mainly on the positive changes in the dynamic processes associated with

563 vertical atmospheric circulations ($\delta H_{Dyn\_v}$). We find that $\delta H_{Dyn\_v}$ patterns correspond

564 well to vertical velocity change patterns at the middle and lower troposphere.

565 Anomalous ascent is primarily triggered by the warming in the middle and lower

566 troposphere over north China in summer and in most of Asia in the other seasons.

567 However, there is anomalous descent in southern China and central India in summer,

568 which is a result of cool anomalies in the middle troposphere due to the reduced latent





heat release caused by the substantial decrease in precipitation. The difference in
diabatic heating at the middle and lower troposphere is related to the difference in
spatial distributions of AOD in the different seasons.
It is well known that the EASM and SASM underwent weakening trends during
the second half of the 20[th] century (Wang et al., 2001; Bollasina et al., 2011). Although
the variations of ASM have been attributed to many factors including internal
variability and external forcing, the strong increases in BC emissions from East and
South Asia (Lund et al., 2019) could play a role in weakening the ASM over the past
decades according to this study. The increased BC also could alleviate the enhanced
precipitation over south China due to GHG increase since the mid-1990s (Tian et al.,
2018). Since the early 2010s, anthropogenic aerosols (including BC and sulfate) have
been decreasing in East China, while they have continued to rise in India; trends which
are expected to continue over the coming decades (Lund et al., 2019; Samset et al.,
2019). Hence, there is a new dipole pattern characterized by decreasing aerosols over
East China and increasing aerosols over India. Given that responses to Asian BC forcing
are linear regionally in the SyRAP-FORTE2 simulations, the impacts of the dipole
pattern on the Asian climate can be roughly estimated by the sum of responses to BC
over China multiplied by -1 and responses to BC over India. The result shows that there
are warm anomalies in north China and cold anomalies in south China, southeast Asia
and most of India, and positive precipitation anomalies over most of China (especially
south China) and southeast Asia, and negative anomalies over India (Fig. S18). It is
overall consistent with the result in Xiang et al. (2023), although their result involves
the combined effect of BC and sulfate.
Large differences in the magnitude and spatial pattern of precipitation responses
to BC can be across models. The smaller precipitation responses over India in FORTE2
relative to PDRMIP may partly be due to much larger BC perturbation in PDRMIP. Liu
et al. (2024) have proposed that the responses of Asian summer rainfall to Asian
aerosols are strongly modulated by regional precipitation biases. Some other factors or
mechanisms may play a role in causing differences in responses to Asian BC, such as
bias of atmospheric circulations, land-atmosphere interaction. Further work to



understand the mechanisms behind model differences in the response to BC would help
to reduce uncertainties and would improve the confidence in future Asian climate
change projections.
**Acknowledgements.** This work is supported by the Second Tibetan Plateau Scientific
Expedition and Research Program (2019QZKK010203). Some of the research
presented in this paper was carried out on the High Performance Computing Cluster
supported by the Research and Specialist Computing Support service at the University
of East Anglia. We acknowledge the Center for Advanced Study in Oslo, Norway that
funded and hosted our HETCLIF centre during the academic year of 2023/24. F. L. is
supported by the Scientific Research of Chengdu University of Information Technology
(Grant No. KYTZ202210). B. H. S., C. W. S., L. J. W., M. J. and R. J. A were supported
by the Research Council of Norway [Grant 324182 (CA3THY)].
**Data availability.** The NOAA-CIRES-DOE 20th Century Reanalysis V3 (20CR)
datasets are obtained from https://psl.noaa.gov/data/gridded/data.20thC_ReanV3.html.
The HadSLP2r is provided by the UK Met Office Hadley Centre and can be
downloaded from at http://www.metoffice.gov.uk/hadobs/hadslp2/. The PDRMIP data
can be accessed through the World Data Center for Climate (WDCC) data server at
https://doi.org/10.26050/WDCC/PDRMIP_2012-2021. Data of the SyRAP-FORTE2
experiments reported in this paper are available without restriction on reasonable
request from Camilla W. Stjern at CICERO Center for International Climate Research.
**Author contribution.** F. L. and B. H. S. designed the study. C. W. S., L. J. W., M. J.
ran the model simulations. F. L. carried out the analysis and visualized the results. All
authors discussed the results and edited the paper.
**Competing interests.** L. J. W. is a member of the editorial board of *Atmospheric*
*Chemistry and Physics*.

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



**Table 1.** Summary of SyRAP-FORTE2 simulations used in the study

| Experiment[a] | Name | Aerosol | Region | GHG | Years |
|---|---|---|---|---|---|
| Baseline | piC | No aerosol | -- | Preindustrial climate conditions (280 ppmv) | 200 |
| Perturbation | BC_CHI | Added BC[b] | East China (95-133°E, 20-53°N) | | |
| | BC_IND | | India (65-95°E, 5-35°N) | | |
| | BC_CHI+IND | | both East China and India region | | |

a. only ARI effect is considered
b. CAMSRA monthly climatology of BC AOD for 2003-2021

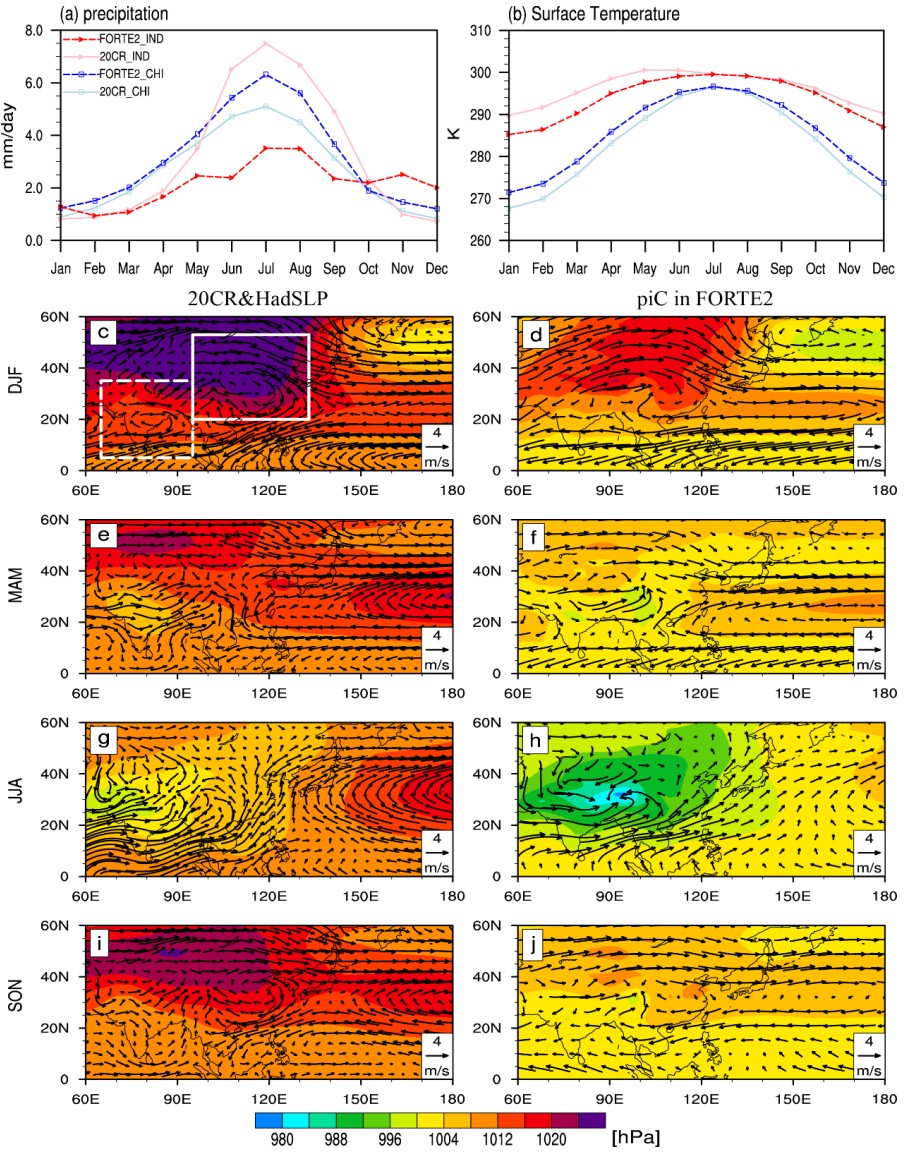

**Figure 1.** Seasonal evolutions of (a) the regional mean precipitation (unit: mm/day) of
20CR (solid lines) and the baseline simulation of FORTE2 (dashed lines) for East Asia
(95°E-133°E, 20°N-53°N, the solid, white box in (c)) (light blue and blue lines), and
India (65°E-95°E, 5°N-35°N, the dashed, white box in (c)) (pink and red lines). (b)
same as (a), but for surface temperature (unit: K). Climate state of SLP (unit: hPa) and
850 hPa horizontal winds (unit: m/s) in (left) 20CR and HadSLP and (right) the baseline
simulation of FORTE2 in four seasons (c-j).

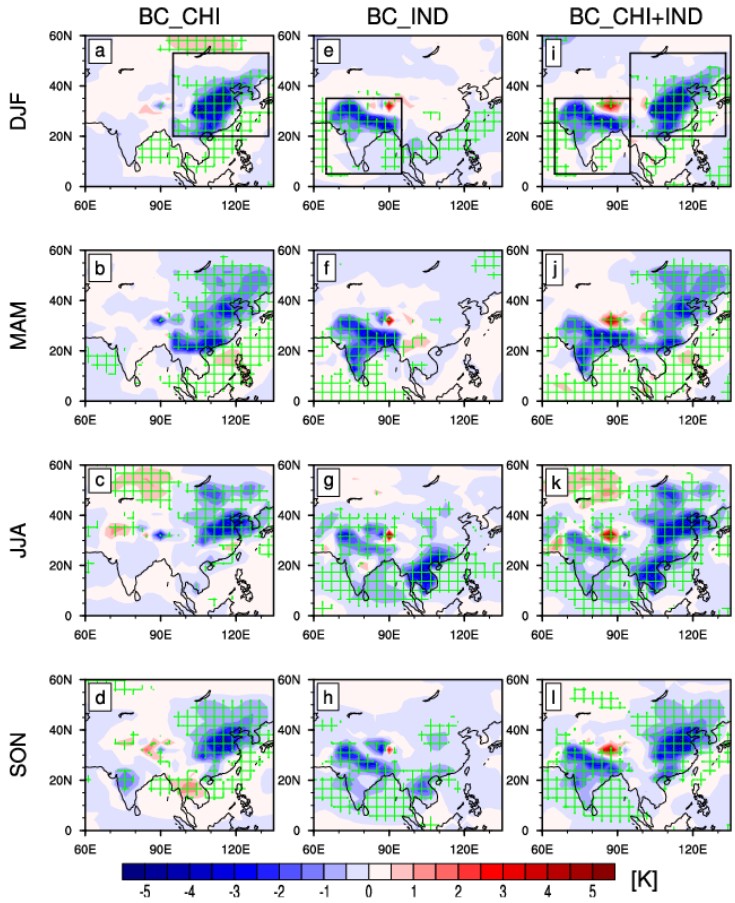

**Figure 2.** Spatial patterns of Ts responses in (a-d) BC_CHI, (e-h) BC_IND, and (i-l)
BC_CHI+IND for four seasons. The green gridlines indicate the regions where the
responses are statistically significant above 95% level based on a two-tailed Student's
t-test. The black boxes in (a), (e) and (i) highlight the region where BC aerosols are
perturbed. Unit: K

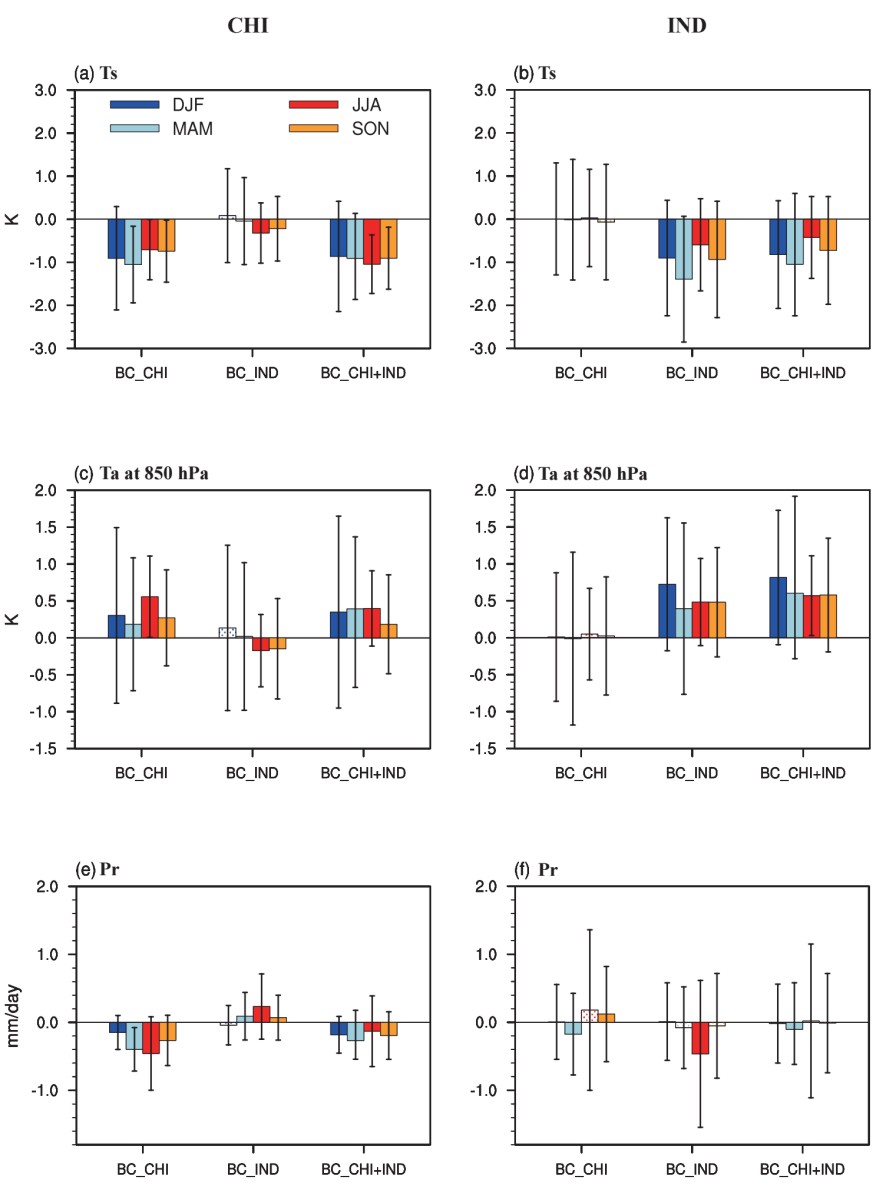

**Figure 3.** Area-averaged land responses of (a-b) Ts, (c-d) Ta at 850 hPa, and (e-f) precipitation over East China (CHI: 95°E-133°E, 20°N-53°N) and India (IND: 65°E-95°E, 5°N-35°N) for four seasons (DJF: blue bars, MAM: light blue bars, JJA: red bars, and SON: yellow bars). Solid bars indicate the responses are statistically significant above 95% level based on a two-tailed Student's t-test. Error bars represent ±1 standard deviations of the response.

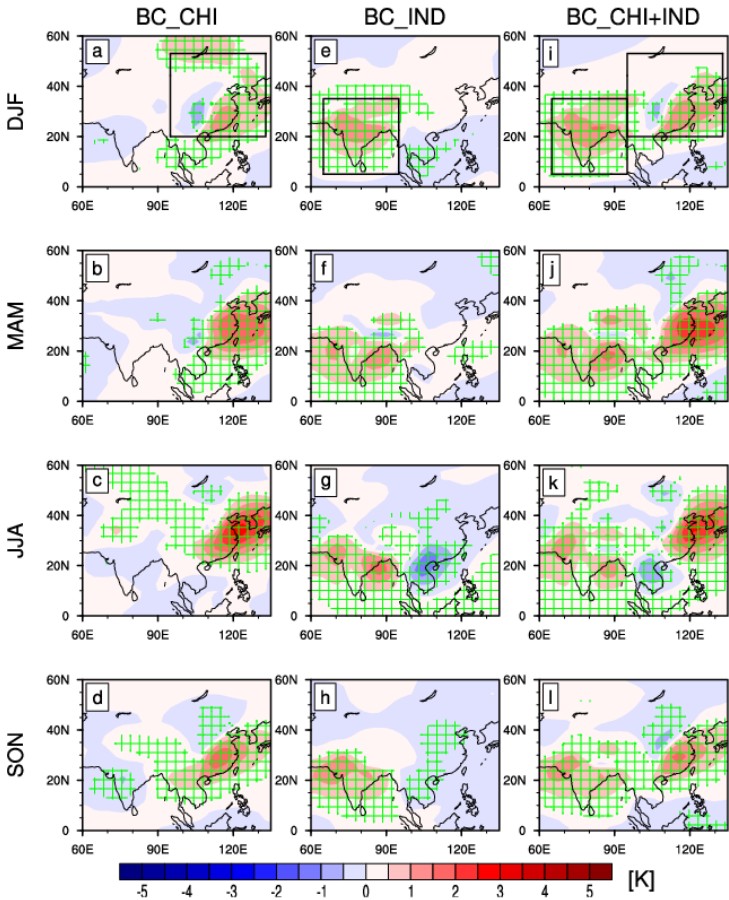

**Figure 4.** Spatial patterns of Ta responses at 850 hPa in (a-d) BC_CHI, (e-h) BC_IND, and (i-l) BC_CHI+IND for four seasons. The green gridlines indicate the regions where the responses are statistically significant above 95% level based on a two-tailed Student's t-test. The black boxes in (a), (e) and (i) highlight the region where BC aerosols are perturbed. Unit: K

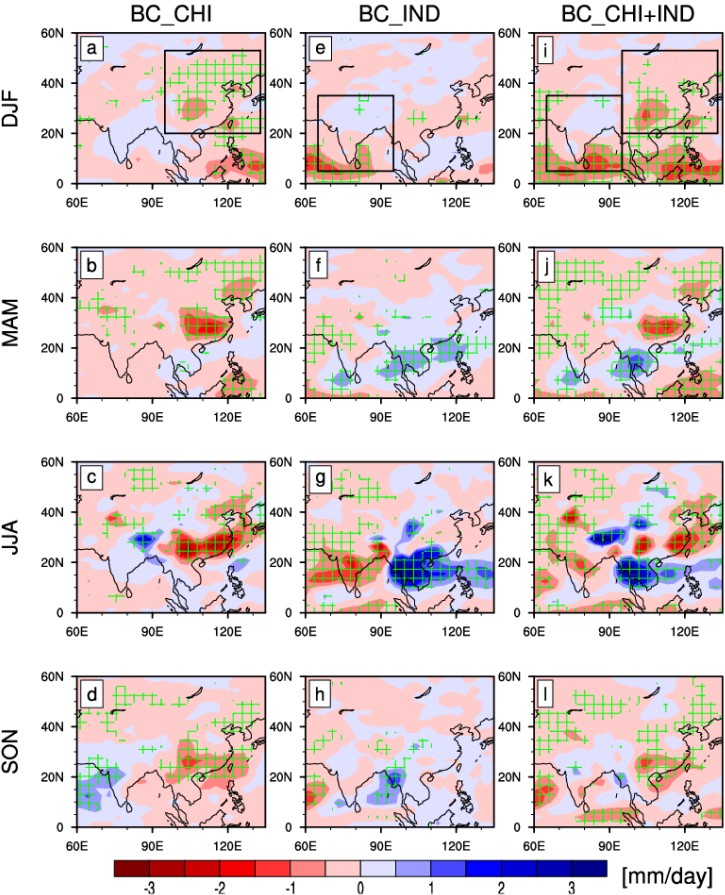

**Figure 5.** Spatial patterns of precipitation responses in (a-d) BC_CHI, (e-h) BC_IND, and (i-l) BC_CHI+IND for four seasons. The green gridlines indicate the regions where the responses are statistically significant above 95% level based on a two-tailed Student's t-test. The black boxes in (a), (e) and (i) highlight the region where BC aerosols are perturbed. Unit: mm/day

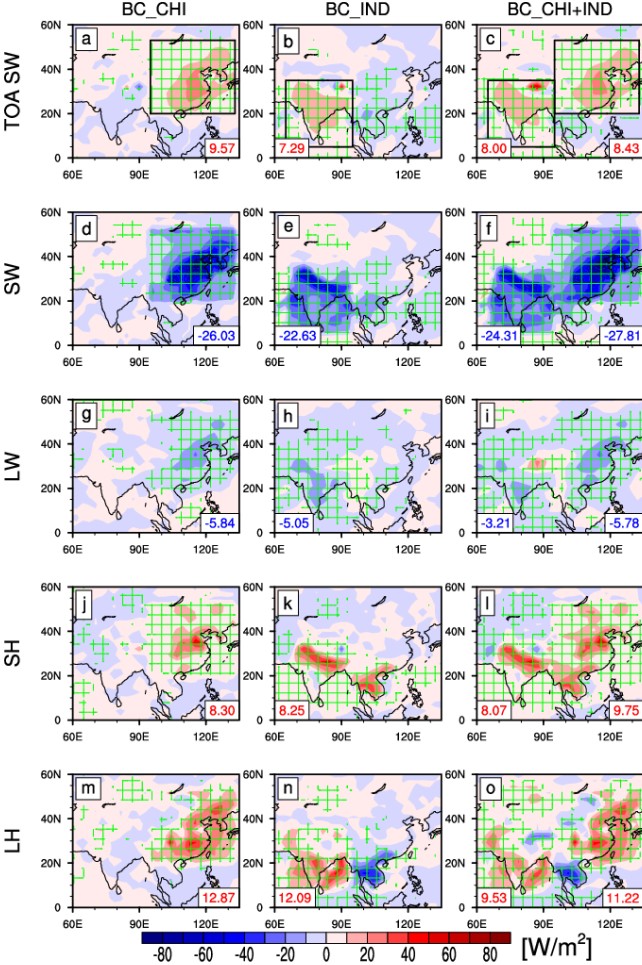

**Figure 6.** Spatial patterns of net TOA and surface energy responses in summer in BC_CHI, BC_IND, and BC_CHI+IND, respectively. (a-c) TOA SW, (d-f) surface SW, (g-i) surface LW, (j-l) surface SH, and (m-o) surface LH. Positive values mean downward for radiation and flux changes. Area-averaged values over East China and India are given in the lower right corners and lower left corners, respectively. The green gridlines indicate the regions where the responses are statistically significant above 95% level based on a two-tailed Student's t-test. The black squares highlight the regions where BC are perturbed. Units: W/m$^2$





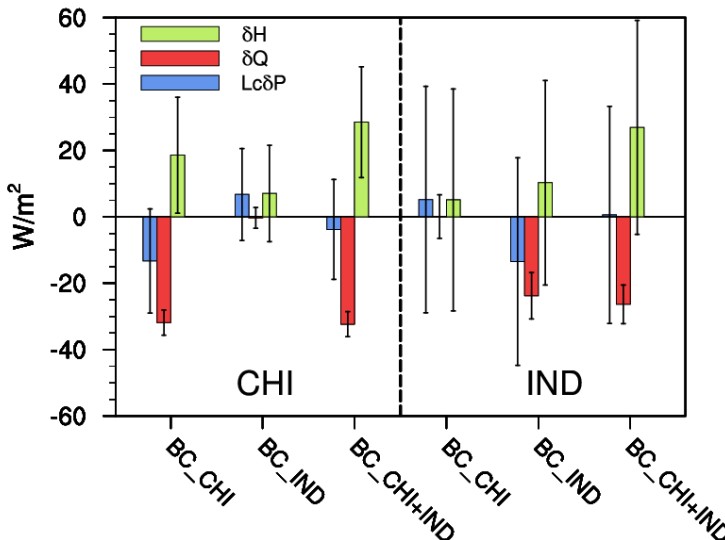

869

**Figure 7.** Summer area-averaged responses of the atmospheric energy budget terms over East China (CHI: 95°E-133°E, 20°N-53°N) and India (IND: 65°E-95°E, 5°N-35°N) in BC_CHI, BC_IND, and BC_CHI+IND. Error bars represent ±1 standard deviations of the response. Unit: W/m$^2$

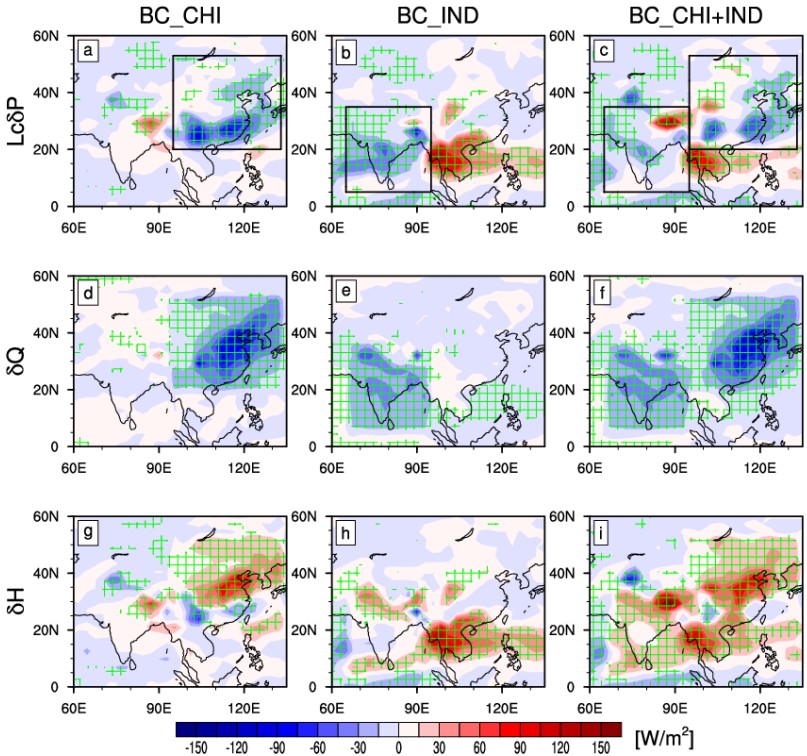

**Figure 8.** Summer spatial patterns of responses of the atmospheric energy budget terms in BC_CHI, BC_IND, and BC_CHI+IND. (a-c) $L_c\delta P$ , (d-f) $\delta Q$ and (g-i) $\delta H$. The green gridlines indicate the regions where the responses are statistically significant above 95% level based on a two-tailed Student's t-test. The black squares highlight the regions where BC are perturbed. Unit: $W/m^2$

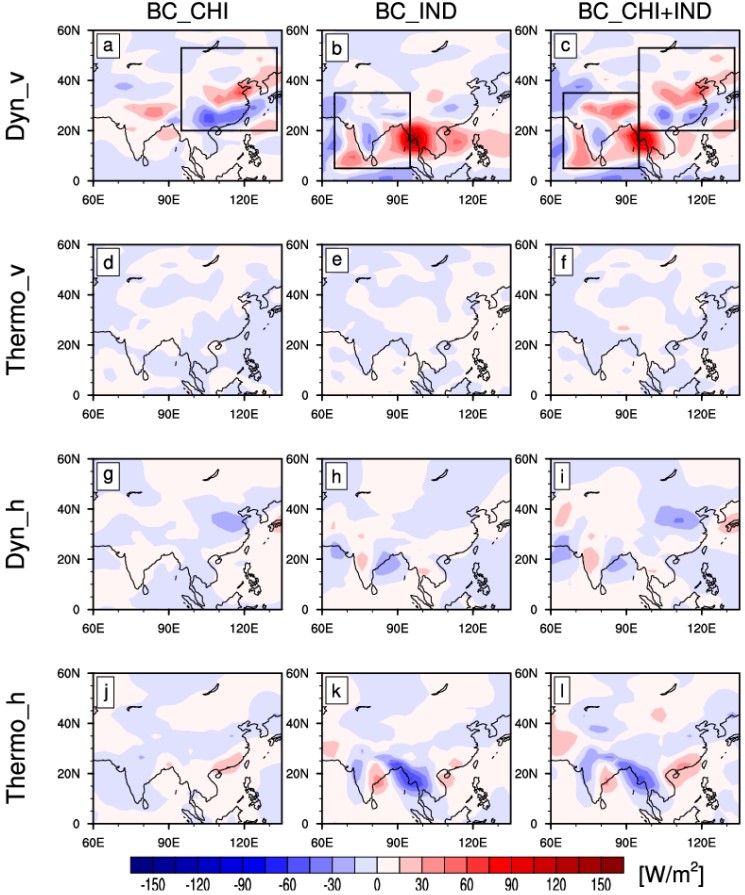

880

**Figure 9.** Summer spatial patterns of responses in the four terms decomposed by $\delta H_m$
in BC_CHI, BC_IND, and BC_CHI+IND. (a-c) the dynamic components with changes
in vertical atmospheric circulations ($\delta H_{Dyn\_v}$), (d-f) the thermodynamic components
with changes in vertical atmospheric circulations ($\delta H_{Thermo\_v}$), (g-i) dynamic
components with changes in horizontal DSE gradients ($\delta H_{Dyn\_h}$), and (j-l)
thermodynamic components with changes in horizontal DSE gradients ($\delta H_{Thermo\_h}$)
The black squares highlight the regions where BC are perturbed. Unit: W/m$^2$

888

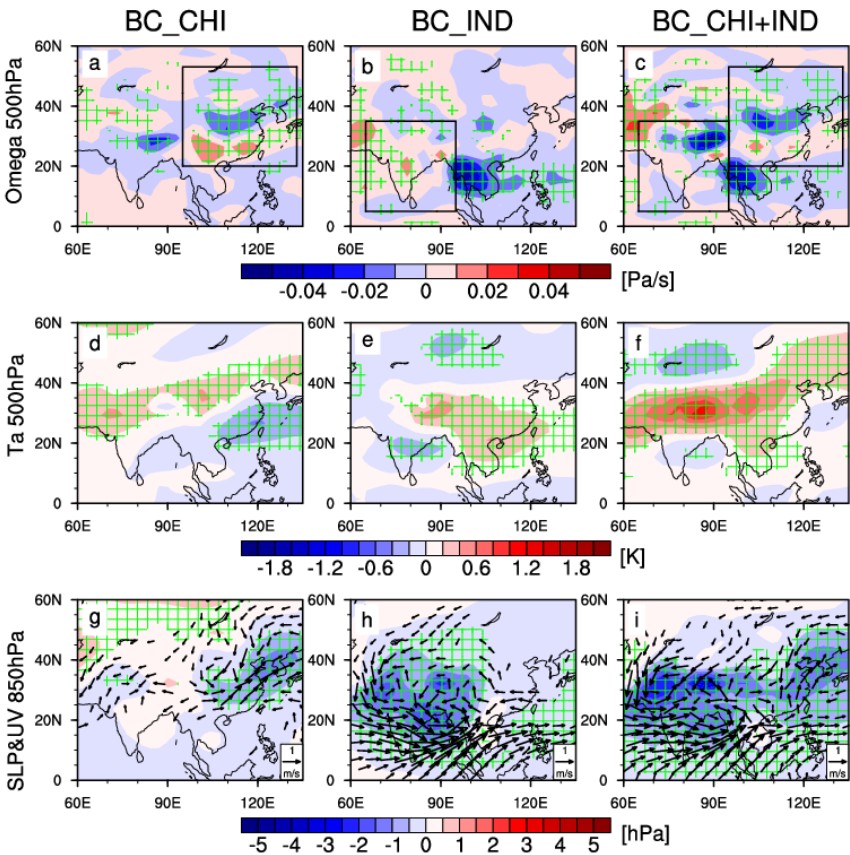

889

**Figure 10.** Summer spatial patterns of responses in (a-c) Omega at 500 hPa (Unit: Pa/s),
(d-f) Ta at 500 hPa (Unit: K), and (g-i) SLP (Unit: hPa) and horizontal wind at 850 hPa
(Unit: m/s) in BC_CHI, BC_IND and BC_CHI+IND. The green gridlines indicate the
regions where the responses are statistically significant above 95% level based on a
two-tailed Student's t-test. Wind vectors are only shown for grid boxes where at least
one component of the wind significant above the 95% level are shown. The black
squares highlight the regions where BC are perturbed.

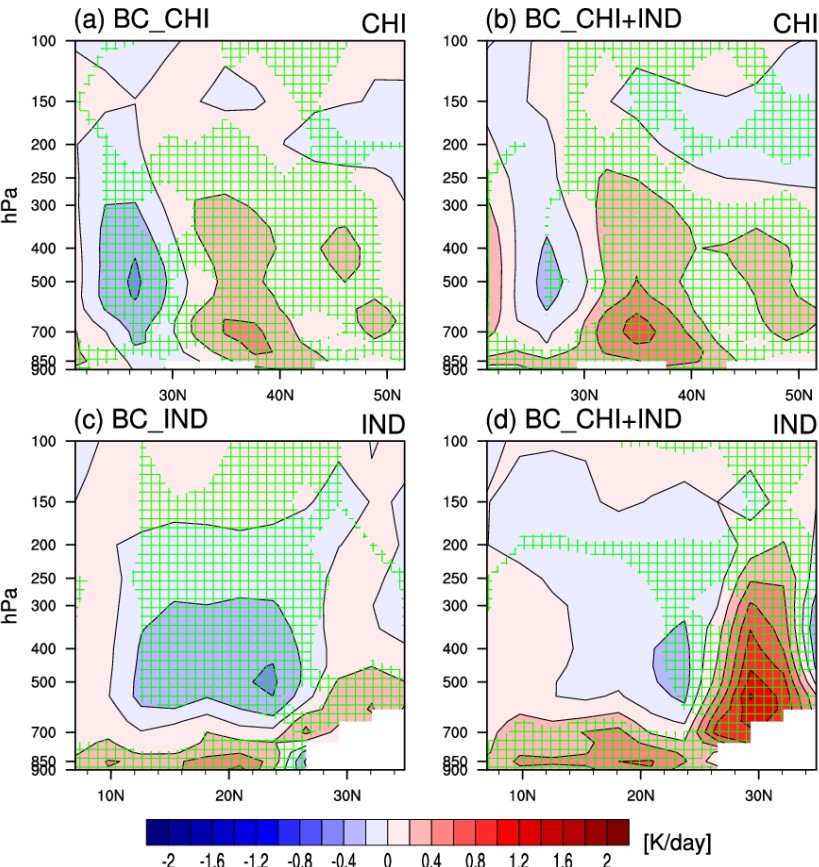

897

**Figure 11.** Zonal mean of diabatic heating responses averaged over (a-b) East China

(95°E-133°E, the black square in Fig.10a) for BC_CHI and BC_CHI+IND, and over

(c-d) India (65°E-95°E, the black square in Fig.10b) for BC_IND and BC_CHI+IND

in summer. The green gridlines indicate the regions where the responses are statistically

significant above 95% level based on a two-tailed Student's t-test. The white part

indicates a symbol of topography. Unit: K/day

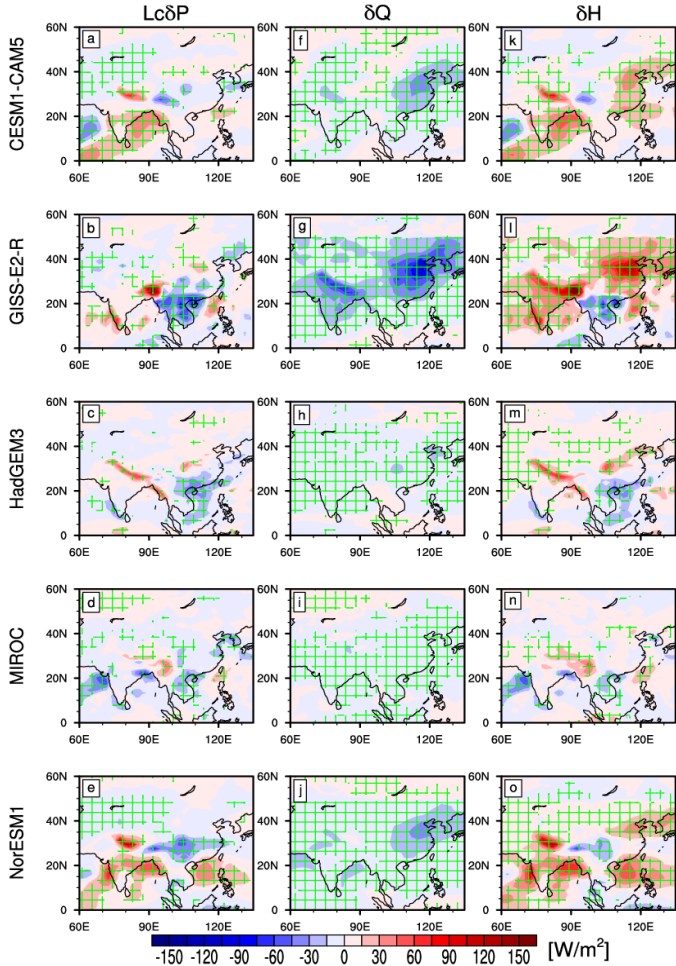

904

**Figure 12.** Summer spatial patterns of responses of the atmospheric energy budget
terms in the five PDRMIP models. (a-e) $L_c\delta P$ , (f-j) $\delta Q$ and (k-o) $\delta H$. The green
gridlines indicate the regions where the responses are statistically significant above 95%
level based on a two-tailed Student's t-test. Unit: $W/m^2$



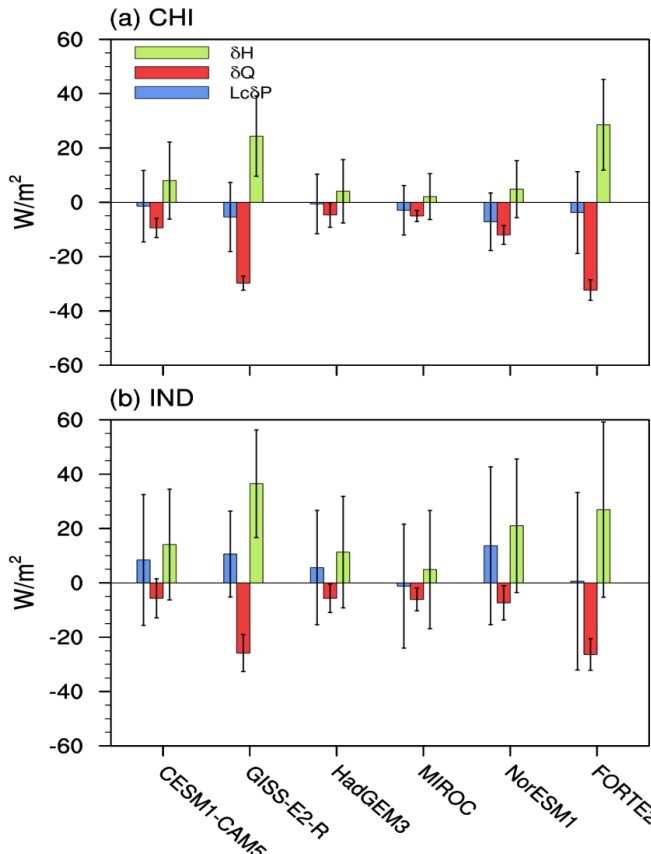

909

**Figure 13.** Summer area-averaged responses of the atmospheric energy budget terms over (a) East China (CHI: 95°E-133°E, 20°N-53°N) and (b) India (IND: 65°E-95°E, 5°N-35°N) in the five PDRMIP models and the BC_CHI+IND simulation in FORTE2. Error bars represent ±1 standard deviations of the response. Unit: W/m$^2$