# Peer review of "Physical processes influencing the Asian climate due to black carbon emission over East and South Asia"

_EGUsphere, 2024_

## Author Response (AR1)

**Referee #1**

This manuscript analyzes the regional responses to Asian black carbon perturbations using a reduced complexity climate model, SyRAP-FORTE2. The authors find a regional response of surface cooling, low-level warming, and precipitation reduction in all seasons. They apply the atmospheric energy budget analysis to explain the mechanisms behind summer precipitation responses. The analysis links the regional precipitation reduction to atmospheric warming induced by BC shortwave absorption, while changes in dry static energy flux divergence partly offset the responses.

The topic is well within the scope of the journal. I think the detailed investigation of the seasonality and the mechanistic insights in this study would be of interest to the relevant communities. I believe this study could be accepted for publication after addressing the following issues.

Thank you for your time and valuable comments on our manuscript. These comments were insightful and have significantly helped improve the manuscript. We have carefully addressed each comment point-by-point. We hope these meet with your approval. The main changes in the manuscript and the responses to the comments are as follows:

**General comments**

The Introduction needs to be better organized. In its current form, the authors list a large number of relevant literature (L61–126) in a somewhat scattered manner. I think more efforts should be made to structure these references logically and to concisely highlight the most pertinent findings. This will help clarify the gap the study aims to fill. On a related note, the Conclusion and Discussion section should better articulate why a reduced-complexity model approach is valuable in this context and how it complements existing work using GCMs or other modeling frameworks. What does the reduced complexity model allow us to see that might be more difficult to isolate in fully coupled

GCMs?

**Response:** Thank you for pointing out these issues. According to your suggestions, we have revised the relevant sections. The updated introduction can be found in Lines 61-101 of the revised manuscript.

For the Conclusion and Discussion section, please refer to Lines 575-583: "Comparative analysis with the CGCMs/ESMs results from PDRMIP has elucidated the key physical mechanisms of Asian climate responses to regional BC perturbations. The thermodynamic process dominates the precipitation reduction, while the dynamic process provides partial compensation. The SyRAP-FORTE2 experiment series allows systematic comparison of impacts of different Asian subregions, aerosol species, and climate backgrounds within a consistent modeling framework. Notably, FORTE2 includes a parameterization of ACI, enabling direct comparison of the relative contributions of ARI and ACI. These related works will be conducted in next step, and may provide new insights into regional aerosol impacts."

In the atmospheric energy budget analysis, the DSE flux divergence is decomposed to mean and eddy terms. However, the eddy component is not discussed in the main text. If the eddy component is negligible, please clarify and justify why it is omitted. Otherwise, consider analyzing how eddies influence the results.

**Response:** We apologize for any lack of clarity regarding this matter. The absence of eddy component analysis in this study was because of computational constraints rather than oversight. We have clarified this. Please refer to lines 399-402: "Due to storage constraints, 3D atmospheric output from FORTE2 was archived on three pressure levels, 250 hPa, 500 hPa, and 850hPa, to capture key aspects of the tropospheric circulation response. While this precludes quantitative analysis of the component terms of $H_m$ and $H_{trans}$, it is sufficient to identify the main contributor to $H_m$."

The description of the model and experiments needs some clarification. I'm confused by which aerosol effects are included in the simulations. L178 indicates the model

includes the semi-direct effect of BC, while L207 seems to suggest otherwise. This should be further discussed in the Conclusion and Discussion. It will help readers gauge how directly they can compare your results with those from more comprehensive aerosol-forcing experiments. Additionally, from my reading, the simulations only use BC, why mention the parameterization of aerosol-cloud interactions for scattering aerosol (L174)?

**Response:** Thank you for the comments. We have removed the original statements in line 174 and 178, and revised the content. Please refer to lines 186-189: "Given that BC primarily influences climate through direct scattering and absorption of radiation (Bond et al., 2013), only climate impacts due to ARI (including the semi-direct effect of BC) were considered here."

In addition, as suggested, we have added a brief discussion in the Conclusion and Discussion. Please refer to lines 587-590: "The FORTE2 simulation only accounts for ARI effects, incorporating the semi-direct effect of BC, while PDRMIP models exhibit varying treatments of indirect effects. The models with all aerosol indirect effects (particularly CESM1-CAM5 and NorESM1) increase precipitation over India, contrasting with the reduced precipitation in FORTE2."

L509–511: I don't find these results very convincing. Indian precipitation response in CESM1 and NorESM1 exhibits opposite sign compared to that in FORTE2. GISS and MIROC show evident precipitation decreases over Southeast Asia, while FORTE2 shows large increases.

**Response:** Thank you for raising this point. We have rephrased the relevant statements in the revised manuscript. Please refer to lines 487-493: "In addition, GISS-E2-R and MIROC show evident decreases in $\delta H$ and precipitation in Southeast Asia, while the other three models show no significant changes. This is contrary to the significant increases seen in FORTE2. There are large differences in the total $L_c\delta P$ responses across these models (Fig. 12a-f). However, some precipitation changes are consistent

in most of the models, such as decreases over South China and increases over North China and northern India."

I think it would be helpful to show the spatial pattern and vertical cross-section of the BC forcing in the supplementary. It might help to understand the seasonality of the BC effects.

**Response:** Thank you for the suggestion. The spatial pattern of the BC AOD has been included in the supplementary as Fig. S1. We have also added clarification regarding the vertical distribution of BC AOD in SyRAP-FORTE2. Please refer to lines 168-175: "Aerosols were distributed vertically uniformly from the second lowest model layer ($\sigma$, or $p/p_{surface}$ = 0.88 or approximately 950 m above the surface) up to a pressure level $p_{min}$. For each gridbox, $p_{min}$ was derived from CAMSRA as either 850 hPa or the lowest pressure level where the 2003-2021 mean BC+OC+SO$_4$ mixing ratio falls below $5 \times 10^9$ kg/kg, whichever value is lower. In topographic regions, additional constraints require $\sigma_{min} < 0.75$ and $p_{min} > 300$ hPa. The seasonal changes of the regional mean $p_{min}$ over China and India are illustrated in Figure S2."

[Figure]

**Figure S1.** Spatial patterns of seasonal AOD of BC within the China (CHI, solid) and the India (IND, dashed).

[Figure]

**Figure S2.** Seasonal evolutions of the regional mean $p_{min}$ for China (CHI, the solid box in Fig. S1) (red line), and India (IND, the dashed box Fig. S1) (blue line).

Careful proofreading is required. There are a number of grammatical and clarity issues throughout the text (e.g., L124, 136, 235, 245, etc.).

**Response:** We have conducted thorough grammatical revisions throughout the manuscript.

L124->L91: "BC aerosol also can impact on …"-> "BC aerosol can also affect …"

L136->L111: "…mean that understanding the causes of differences between studies is very difficult."-> "…make it challenging to understand the causes of differences between studies."

L235->L217: "To evaluate the ability of FORTE2 to represent the observed climate monthly…"-> "To evaluate the skill of FORTE2 to simulate observed climate variables…"

L245->L227: "…modern aerosol concentrations/emissions and greenhouse gases with the year 2000"-> "modern aerosol concentrations/emissions and greenhouse gases under year-2000 conditions"

**Individual comments**

L88: clarify what is "enhanced upper-level atmospheric temperature"

**Response:** It has been revised. Please refer to lines 75-78: "Xie et al. (2020) have proposed that the precipitation responses mainly result from the upper-level atmospheric heating over Asia, which enhances the upper-level meridional land-sea thermal gradient and subsequently strengthens the low-level monsoon circulation."

L149: referring to "emissions" may be misleading when BC AOD is prescribed in the models.

**Response:** L134: "emissions"-> "perturbations"

L170: do you mean to state that FORTE2 has good skills?

**Response:** Yes, it has been clarified. Please refer to lines 156-158: "Blaker et al. (2021) have thoroughly evaluated FORTE2's skill in simulating the atmosphere, ocean and major climatic modes, suggesting that FORTE2 can satisfactorily simulate a climate state and climate variability."

L204: what are "different background climate states"?

**Response:** It has been clarified. Please refer to line 182: "between the pre-industrial and present-day climate conditions"

L238: It does not seem right.

**Response:** It has been modified. L220: spanning 1806 to 2015

L367: please be explicit about what vertical temperature differences reduce SH and how heating reduces LH.

**Response:** It has been clarified. Please refer to lines 349-352: "The reduced SH is caused by the weakened vertical temperature differences between the surface and lower atmosphere. The decrease in upward LH is associated with upper-level heating through atmospheric stabilization that suppresses moisture transport, and reduced soil moisture availability due to decreased rainfall."

L547: precipitation increases in Southeast Asia.

**Response:** It has been modified accordingly. Please refer to line L530.

Figure 12: suggest to include the FORTE2 results to facilitate a direct comparison.

**Response:** The figure has been replotted.

[Figure]

**Figure 12.** Summer spatial patterns of responses of the atmospheric energy budget terms in the five PDRMIP models and the BC_CHI+IND simulation in FORTE2. (a-e) $L_c\delta P$ , (f-j) $\delta Q$ and (k-o) $\delta H$. The green gridlines indicate the regions where the responses are statistically significant above 95% level based on a two-tailed Student's t-test. Unit: W/m$^2$

**Referee #3**

The authors investigate the response of Asian monsoon regions to South and East Asian BC perturbations using a simplified climate model. They analyze the seasonal response to BC forcing, with a focus on summertime for the energy budget analysis. This study compares the relative effects of local versus remote forcing, as well as their combined impact. The work underscores the importance of the spatial location of aerosol forcing in determining regional climate responses. I have several concerns that I would like the authors to address.

We sincerely appreciate your constructive feedback, which has greatly improved our manuscript. We have carefully addressed all comments through comprehensive revisions. We hope these meet with your approval. The main changes in the manuscript and the responses to the comments are as follows:

1. The Introduction could be better structured to more clearly articulate the motivation for using a simplified climate model to investigate the impacts of regional aerosol emissions. Given the extensive body of literature on aerosol-driven responses in the Asian monsoon region, the referenced studies cloud be organized to highlight the value of using a simplified model and specifically using observed AOD over the past two decades as forcing. However, the aerosol forcing used requires further clarification (see point 2 below).

Response: Thank you for your valuable comments. We have revised the Introduction accordingly. Please refer to lines 61-130.

2. The description of experimental design is confusing. Some information needs clarification.

● The method of "adding" a single aerosol species is not well explained. Is the prescribed AOD distribution from reanalysis data considered the "added" aerosol forcing? Is monthly mean AOD from reanalysis repeated each year in the experiment? Is there a seasonal cycle of AOD in the experiments that could influence the seasonal

response to BC? A clearer description of the regional AOD used, along with a figure showing its spatial distribution and trends, would be helpful.

**Response:** We appreciate this valuable comment. The relevant descriptions have been clarified in the revised manuscript (Lines 163-167): "The CAMSRA incorporates anthropogenic BC emissions from the MACCity inventory (Granier et al., 2011) for 2003-2010, transitioning to Representative Concentration Pathway 8.5 emissions (Riahi et al., 2011) post-2010. The simulations were idealized with monthly AOD climatologies prescribed as repeating annual cycles."

The spatial pattern of the BC AOD has been included in the Supplementary Fig. S1.

[Figure]

**Figure S1.** Spatial patterns of seasonal AOD of BC within the China (CHI, solid) and the India (IND, dashed).

● Lines 180–187 mention aerosol-radiation interactions (ARI) versus aerosol-cloud interactions (ACI), while line 171 refers to the inclusion of the semi-direct radiative effect. Which of these radiative effects are actually included in the experiments in this work? Moreover, described experiments aren't directly analyzed or discussed in this work.

**Response:** Apologize for any confusion caused. This section (L180-187) has been relocated to the introduction with a brief discussion. Please refer to lines 127-130: "The SyRAP-FORTE2 framework enables comprehensive analysis of climate effects of different regional aerosol perturbations and aerosol species, as well as ARI versus ACI, and allows comparison of their relative importance and interactions (Stjern et al., 2024)."

The original statement regarding "The semi-direct effect of BC is included" has been

removed. We have rephrased the relevant statements. Please refer to lines 186-189: "Given that BC primarily influences climate through direct scattering and absorption of radiation (Bond et al., 2013), only climate impacts due to ARI (including the semi-direct effect of BC) were considered here."

● The parameterization discussed in lines 173–178 and does not appear to be applied in the current study. If this is the case, why is it included?

**Response:** This section has been removed.

3. There are substantial differences between the response and patterns in the energy analysis based on SyRAP-FORTE2 BC_CHI+IND simulations and those in PDRMIP models. In PDRMIP, aerosol emissions are artificially amplified, however, responses in SyRAP-FORTE2 have a similar magnitude to those from PDRMIP experiments, for example, $\delta Q$ in GISS-E2-R. Additionally, the radiative effects included across PDRMIP models vary. Given these discrepancies in forcing magnitude, model configuration, and included radiative effects, I question the robustness of the conclusion that "the results of these models are overall consistent qualitatively".

**Response:** We fully agree with you that substantial differences exist between FORTE2 and PDRMIP simulations including model configurations, aerosol forcing magnitudes, and responses in intensity and spatial patterns. However, the "qualitative consistency" conclusion specifically indicates that: thermodynamic processes ($\delta Q$) dominantly decrease precipitation, while dynamic processes ($\delta H$) increase precipitation and partially offset the effects of $\delta Q$. We have clarified this in the revised manuscript. Please refer to lines 511-516.

In addition, we have addressed a brief discussion regarding the response magnitude in the Conclusion and Discussion section, please refer to lines 591-594: "Additionally, despite the much weaker BC forcing in FORTE2, it produces larger thermodynamic ($\delta Q$) and dynamic ($\delta H$) responses than most PDRMIP models (except for GISS-E2-R).

This may arise from the absence of wet deposition feedbacks in FORTE2 (Stjern et al., 2024).”